# Random Registers for Cross-Domain Few-Shot Learning

**Shuai Yi** [1 2]  **Yixiong Zou**[✉ 1]  **Yuhua Li**[✉ 1]  **Ruixuan Li**[✉ 1]

## Abstract

Cross-domain few-shot learning (CDFSL) aims to transfer knowledge from a data-sufficient source domain to data-scarce target domains. Although Vision Transformer (ViT) has shown superior capability in many vision tasks, its transferability against huge domain gaps in CDFSL is still under-explored. In this paper, we find an intriguing phenomenon: during the source-domain training, prompt tuning, as a common way to train ViT, could be harmful for the generalization of ViT in target domains, but setting them to random noises (i.e., random registers) could consistently improve target-domain performance. We then delve into this phenomenon for an interpretation. We find that learnable prompts capture domain information during the training on the source dataset, which views irrelevant visual patterns as vital cues for recognition. This can be viewed as a kind of overfitting and increases the sharpness of the loss landscapes. In contrast, random registers are essentially a novel way of perturbing attention for the sharpness-aware minimization, which helps the model find a flattened minimum in loss landscapes, increasing the transferability. Based on this phenomenon and interpretation, we further propose a simple but effective approach for CDFSL to enhance the perturbation on attention maps by adding random registers on the semantic regions of image tokens, improving the effectiveness and efficiency of random registers. Extensive experiments on four benchmarks validate our rationale and state-of-the-art performance. Codes and models are available at https://github.com/shuaiyi308/REAP.

[1]School of Computer Science and Technology, Huazhong University of Science and Technology, Wuhan, China [2]School of Artificial Intelligence and Automation, Huazhong University of Science and Technology, Wuhan, China. Correspondence to: Yixiong Zou <yixiongz@hust.edu.cn>, Yuhua Li <idcliyuhua@hust.edu.cn>, Ruixuan Li <rxli@hust.edu.cn>.

*Proceedings of the 42nd International Conference on Machine Learning*, Vancouver, Canada. PMLR 267, 2025. Copyright 2025 by the author(s).

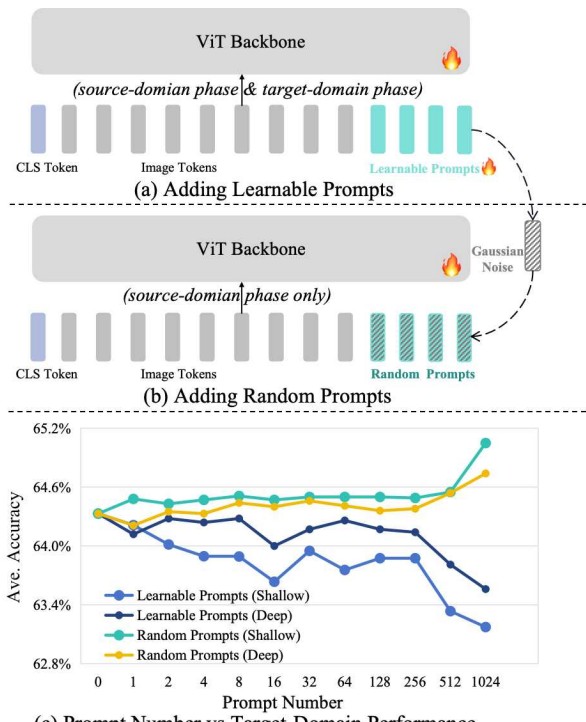

*Figure 1.* (a) Vision Transformer (ViT) takes the CLS token, image tokens, and learnable prompts as input for both the source-domain training and the target-domain testing. (b) We replace the learnable prompts with random noises (i.e., random prompts) on the source domain, which are then dropped on the target-domain phase. (c) We find an intriguing phenomenon: unlike other tasks, prompts learned on the source domain harm the target-domain performance, while random prompts (both deep and shallow types) improve it.

## 1. Introduction

Large models achieve great success in many tasks due to their power in learning from large-scale datasets (Zou et al., 2022; Chen et al., 2023b), with large vision models majorly taking the VIT-based architectures (Zhang et al., 2022a; Noori et al., 2024; Zou et al., 2024b). However, the generalization of ViT under the extreme cross-domain data-scarce scenarios still needs to be explored since collecting sufficient data for every domain does not always hold in the real world (Tseng et al., 2020; Naseer et al., 2021). To this end, Cross-Domain Few-Shot Learning (CDFSL) has been proposed to transfer general knowledge from the source

domain (e.g., ImageNet (Krizhevsky et al., 2017)) with numerous nature images to target domains (e.g., medical datasets (Codella et al., 2019)) with only a few labeled examples (Guo et al., 2020; Zou et al., 2021). Huge domain gaps between source and target domains make it difficult to transfer source-domain-trained ViT to target domains for few-shot learning (Zou et al., 2024a).

To handle this task, we focus on a common way of training the ViT model: the Visual Prompt Tuning (Jia et al., 2022) method (Fig. 1a), which concatenates additional learnable prompts to the input sequence of ViT. We train the ViT model on the source-domain dataset with learnable prompts and evaluate the target-domain performance through the prototype-based method (Snell et al., 2017). However, we find such a common way of ViT training causes significant performance degradation on the target domain datasets (Fig. 1c). Instead, interestingly, we directly abandon the learning of prompts and randomize these prompts with Gaussian noises (Fig. 1b), and find this operation would consistently improve target-domain performance when increasing the number of prompts, for both the shallow and deep prompts. Surprisingly, the highest performance is achieved when the prompt number reaches the maximum that the GPU memory can hold.

In this paper, we delve into this phenomenon for an interpretation. As shallow prompts show higher performance and are simpler, we mainly target this kind of prompt, and we call it Register following (Darcet et al., 2024). We first find attention maps on target domains always fail to find semantic objects, indicating a poorly transferred attention network of ViT. Then, we measure the transferability of attentions through the sharpness of loss landscapes (Foret et al., 2021; Zou et al., 2024a) against the perturbation on attentions, and quantitatively verify learnable registers decrease the transferability while random registers increase it. Inspired by this, we interpret **random registers as a novel way of perturbing attention to conducting the sharpness-aware minimization** (Foret et al., 2021), therefore improving the transferred attention on target domains. In contrast, we find learnable registers capture domain information during the training on source datasets, represented as viewing irrelevant visual patterns (e.g., background) as important cues for recognition. This can be viewed as a kind of overfitting to the source dataset, thereby increasing the sharpness.

Based on the above interpretations, we further propose a simple but effective method to boost ViT's transferability to target domains, by improving the effectiveness and efficiency of random registers (so that the number of appended random registers can be small). During the source-domain phase, since the core of the random registers is perturbing attention maps by prompts (tokens), we add random registers to semantic regions of the input image tokens, by randomly

replacing clustered image patches with random noises. This operation increases the ratio of perturbed information in the attention maps, which increases the efficiency of random registers. During the target-domain phase, we maintain all tokens as input and append learnable registers as prompts for finetuning to take advantage of their absorption of domain information. Experiments on four datasets validate our rationale for the interpretations and prove that we can outperform state-of-the-art works.

In summary, our contributions can be listed as follows.

- To the best of our knowledge, we are the first to find that prompt learning on the source domain harms the transferability to target domains, but utilizing random registers would consistently improve it.

- We delve into this phenomenon for an interpretation: the learnable registers absorb domain information, represented as the focus on regions irrelevant to recognition, while random registers novelly perturb attention maps for sharpness-aware minimization.

- Based on the interpretation, we propose a novel method to enhance perturbations on attention maps, which adds random registers to semantic regions of image tokens, therefore improving the effectiveness and efficiency of random registers and enhancing model transferability.

- Extensive experiments on four benchmark datasets validate our rationale and state-of-the-art performance.

## 2. Delve into the Registers in ViT-based Cross-Domain Few-Shot Learning

### 2.1. Preliminaries

Cross-Domain Few-Shot Learning (CDFSL) requires the model to learn from a source-domain dataset with sufficient training samples, and then transfer to the downstream tasks, aiming to recognize the target-domain datasets by only a few training data (Wang & Deng, 2021; Li et al., 2022). Specifically, the source-domain and the target-domain datasets are denoted as $D^S = \{I_j^S, y_j^S\}_{j=1}^{N^S}$ with class labels $y \in C^S$, and $D^T = \{I_j^T, y_j^T\}_{j=1}^{N^T}$ with class labels $y \in C^T$ respectively. Source and target classes are disjoint: $C^S \bigcap C^T = \emptyset$, and large domain gaps exist between them. During the training on source domain $D^S$, the goal is to train the whole model by minimizing the cross-entropy loss

$$L = \frac{1}{N} \sum_{j}^{N} L_{cls}(\phi(f(I_j^S)), y_j^S), \qquad (1)$$

where $f(\cdot)$ denotes a feature extractor (e.g., ViT) and $\phi(\cdot)$ denotes a fully connected (FC) layer (i.e., classifier). Then

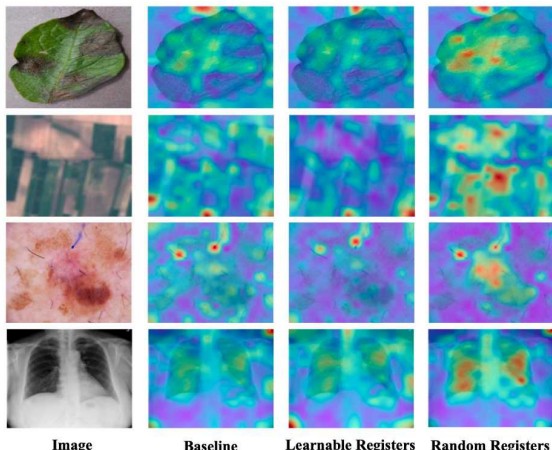

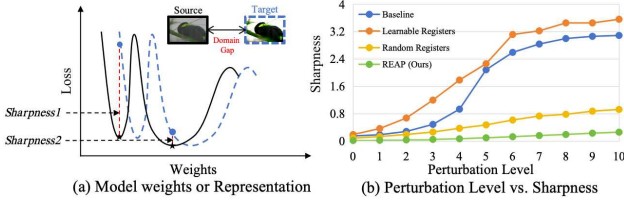

*Figure 3.* (a) When the data is shifted from the training data, the originally effective weights or representations (i.e., minima in the loss landscape) may not be in low loss (blue point), whereas the increase of loss (i.e., sharpness) measures the vulnerability to domain shifts. (b) The model with learned registers consistently shows higher sharpness than others, indicating lower robustness to domain shifts. In contrast, the lower sharpness of random registers verifies that the attention is more transferable across domains.

*Figure 2.* Visualization of the model's attention in the last block, indicating learnable registers make the model unable to recognize semantic regions on target domains. In contrast, random registers effectively guide the model's attention to the object.

$f(\cdot)$ is transferred to target-domain datasets $D^T$, where only 1 or 5 samples are available for each class. Following current works (Chen et al., 2021; Fu et al., 2021), $n$-way $k$-shot episodes are sampled for target-domain training and evaluation. It means each episode consists of a support set $\{I_{hj}^T, y_{hj}^T\}_{h=1,j=1}^{n,k}$ with $n$ classes and $k$ samples in each class for training and a query set $\{I_q^T\}$ for evaluation. The classification is conducted by the distance between class prototypes and samples

$$\hat{y_q^T} = \arg\min_j d(\frac{1}{k}\sum_j f(I_{jh}^T), f(I_q^T)), \quad (2)$$

where $d(\cdot, \cdot)$ denotes the Euclidean distance function. We follow StyAdv (Fu et al., 2023) to use the Vision Transformer (ViT) with the DINO pretrained as our backbone network. Since in Fig. 1 the shallow prompt shows more significant changes in performance, below we mainly target to study this kind of prompt learning[1], which adds learnable tokens to the input token sequence (similar to the [CLS] token). Following (Darcet et al., 2024), we term it **registers**. Then, the ViT takes the [CLS] token, image tokens, and learnable registers as inputs $P$ fed into feature extractor $f(\cdot)$ later, written as

$$f(P) = f(C(T^C, T(I), T^{R_1}, T^{R_2}, \cdots, T^{R_{\tilde{n}}})), \quad (3)$$

where the [CLS] token is denoted as $T^C \in R^d$, image tokens as $T(I) \in R^{n^t \times d}$, and registers as $T^R \in R^d$, $\tilde{n}$ is the register number, $C(\cdot, \cdot)$ means the concatenation of different tokens (Fig. 1a). In the following, we will explore why random registers improve target-domain performance.

---

[1]For its deep version, please refer to the Appendix, which shows similar results to the shallow one.

## 2.2. How do registers affect the attention?

As ViTs are prevailing for their self-attention mechanism, we first visualize the attention map of the model with learnable or random registers on the target domain in Fig. 2. As the CLS token feature from the last ViT block is utilized as the final output of the backbone network, we visualize the attention maps of the CLS token against image tokens in the last block. In visualizations, the red color indicates large attention scores while blue indicates small ones.

From Fig. 2, we can see the baseline model struggles to capture semantic regions when transferred to target domains. Adding learnable registers further intensifies the model's focus on irrelevant regions, while random registers help the model focus on semantic objects in target domains. Therefore, we hypothesize the attention network, as the core of ViT, may not be well transferred to target domains.

To verify our hypothesis, we quantitatively measure the transferability of the attention network by the sharpness of loss landscapes (Foret et al., 2021; Zou et al., 2024a). Specifically, given a well-trained model from the source domain, each weight (Foret et al., 2021) or representation (Zou et al., 2024a) for the source-domain input data can be viewed as a point associated with a loss value, where each minima point represents a good weight or representation (Fig. 3a). Sharpness measures how the loss value changes when the data shifts from the source domain, with lower loss change indicating better robustness to domain shifts.

To evaluate the sharpness of the attention network, we add perturbations to the attention map by Gaussian noises

$$Sharpness = max_\epsilon\left[L(A + \epsilon) - L(A)\right], \quad \epsilon \sim N(0, \sigma) \quad (4)$$

where $A$ denotes the attention map, $\epsilon$ controls the perturbation level. As can be seen in Fig. 3b, all models experience a rise in loss when subjected to these perturbations. However, the model with learnable registers consistently shows a much higher sharpness than others, indicating they push the model to be less robust to domain shifts. The model with

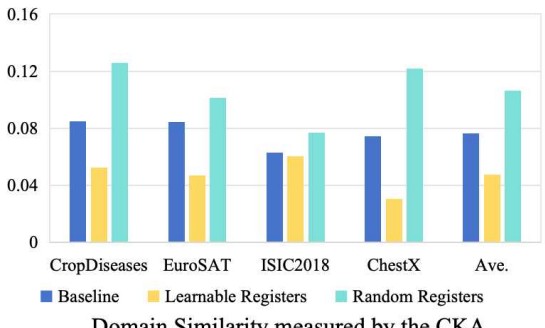

Domain Similarity measured by the CKA

*Figure 4.* Adding learnable registers consistently decreases the domain similarity, indicating registers contain domain information only valid on the source dataset, increasing the sharpness. Meanwhile, random registers consistently improve the domain similarity, demonstrating the model is pushed to learn domain-agnostic information, as is interpreted as sharpness-aware minimization.

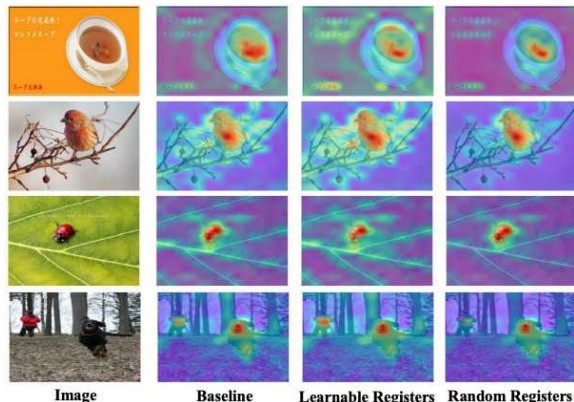

*Figure 5.* The attention on the source dataset. Learnable registers make the model concentrate on regions irrelevant to the object, which can be viewed as a kind of overfitting. In contrast, random registers effectively guide the model's attention to the object.

random registers exhibits a smaller increase than the baseline model, quantitatively verifying that random registers improve the transferability of attention.

## 2.3. Why do registers contribute to transferability?

Based on the above analysis, we interpret random registers as a kind of sharpness-aware minimization (SAM (Foret et al., 2021), please refer to the appendix for more details) as follows. Following (Foret et al., 2021), SAM can be formulated as

$$L_{SAM} = \min_{\omega} \left[ \max_{\|\epsilon\|_2 \leq \rho} L(\omega + \epsilon) \right] + \lambda(\|\omega\|_2^2), \quad (5)$$

where $\omega$ refers to the components vulnerable to data shifts. In contrast, adding random registers to the input sequence can be reflected in the attention maps as

$$A_{i,j} = \frac{e^{Q_i K_j^T}}{\sum_{k=1}^{n} e^{Q_i K_k^T} + \sum_{k=1}^{\tilde{n}} e^{Q_i \bar{K}_k^T}}, \quad (6)$$

where $n$ is the total patch number in an image and $\tilde{n}$ is the number of registers we add to the input sequence fed into ViT. For each image token or CLS token, its query $Q_i$ will be multiplied with a randomized key $\tilde{K} = T^{R_i} W^K$ from random registers $T^{R_i}$. Therefore, the term $\sum_{k=1}^{\tilde{n}_k} e^{Q_i \times \bar{K}_k^T}$ will also be a random noise. Denote such a noise as $\epsilon^R$, we can rewrite SAM in Eq. 5 as

$$L_{SAM} = \min_{\omega} \left[ \max_{\epsilon} L(A + \epsilon^R) \right] + \lambda(\|\omega\|_2^2). \quad (7)$$

As the attention is verified to be vulnerable in Fig. 2, it is reasonable to add noise to the attention map. In all, adding random registers can be regarded as **a novel way to add perturbations to the attention maps** and can be viewed as a kind of SAM, which helps the model to find a more flattened minimum and be well transferred to target domains.

## 2.4. Why do registers influence sharpness?

Then, we delve into why registers influence sharpness and transferability. Since the domain gap is the most important challenge for transferability in CDFSL, we follow (Oh et al., 2022; Davari et al., 2022) to take the CKA (Kornblith et al., 2019) similarity as a tool to measure the domain similarity between source and target datasets. Specifically, after the backbone network is trained on the source datasets, we extract features of images from different domains and then compute the CKA similarity. A higher value means a higher similarity between the source and target domains, representing more domain-agnostic information in the backbone network. As shown in Fig. 4, we can see:

(1) Adding learnable registers trained on the source domain can significantly drop the CKA similarity, indicating the registers contain domain information learned from the source datasets. With the increase of the register number in Fig. 1, more domain information is absorbed by registers, guiding the model to learn more domain-specific information that is only valid on source-domain classification.

(2) Random registers, added as random Gaussian noises, can consistently increase the CKA similarity, and more random registers in Fig. 1 are more beneficial to the model. Since random registers are abandoned during the target-domain phase, this result means random registers push the model to learn domain-agnostic information valid across domains, as is explained as sharpness-aware minimization.

Based on it, **we interpret the increased sharpness as a result of absorbed domain information**, which is only effective for the source domain. Therefore, when the data is shifted from the source domain, such a model cannot extract effective features from the shifted data, leading to a significant increase in the loss, i.e., larger sharpness.

To further understand how the domain information is reflected in the pattern learning, we visualize the attention map of the model with learnable and random registers on the source domain in Fig. 5. We can see that by adding learnable registers, the model captures more regions irrelevant to the recognition of the object (e.g., patterns in the background), while random registers help the model focus more on the object. For learnable registers, this result means the model **takes these irrelevant regions as crucial clues for classification**. This phenomenon essentially represents the model's overfitting to the source domain since these regions are class-irrelevant and can be source-domain-specific, i.e., they may be useful for the source domain but invalid for other domains. For random registers, this result means the model is pushed to **focus less on regions irrelevant to the object**, e.g., patterns outside the object, which are more likely to be class-relevant and transferable across domains. That is, the model is pushed to learn domain-agnostic information, consistent with the analysis above.

## 2.5. Conclusion and Discussion

Based on these, we interpret as follows: By applying learnable registers, the model focuses more on the regions irrelevant to the object recognition. This can be viewed as a kind of overfitting to the source domain, which contains more domain information and increases the sharpness of the loss landscapes, i.e., harms the generalization to target domains. Applying random registers is an efficient and novel way to perturb the attention maps fed into the model and can be viewed as a kind of sharpness-aware minimization. Therefore, the model is trained to find a flattened minimum in loss landscapes and of good transferability to target domains. As a result, the model focuses less on the regions outside the object and generalizes better to target domains.

## 3. Method

Based on the above analysis and interpretation, we further develop a simple but effective approach called Random Registers Enhanced Attention Perturbation (REAP) for the CDFSL task by improving the effectiveness and efficiency of random registers, so as to boost the model's transferability. Meanwhile, according to the characteristics of these two types of registers, we further propose a two-stage training strategy for the model, as shown in Fig. 6.

During the source-domain stage, we design an improved random-register-based method to improve both the effectiveness and efficiency of random registers, which encourages the model to learn a flattened minimum in loss landscapes on the source domain. During the target-domain stage, instead of random registers that perturb domain information, we come back to the regular learnable registers for their absorption of domain information, to help the model adapt

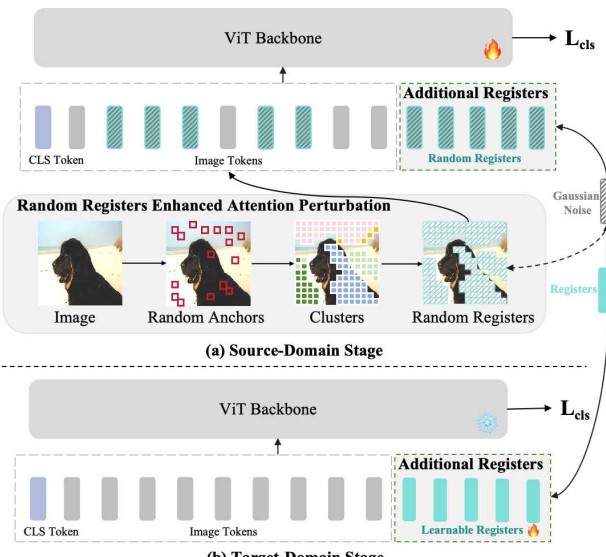

*Figure 6.* Overview of our framework, which consists of two stages. (a) In the source-domain stage, we randomly drop clustered image tokens and replace dropped tokens with random registers (vectors of Gaussian noises), which improves the effectiveness and efficiency of random registers, thereby encouraging the model to learn domain-agnostic information. (b) In the target-domain stage, we activate the additional registers to the learnable state to help the model adapt to target domains with only scarce data.

to target domains with scarce data.

### 3.1. Random Registers Enhanced Attention Perturbation (REAP)

In Fig. 1, the random register can be effective only when the number of it is very large, reaching the limit that the GPU memory can hold, which is ineffective and inefficient.

To handle this problem, we think about the ratio of perturbed information that random registers can bring to the attention map of the model. Specifically, the reason why we need a large number of random registers is that *the attention maps has already captured rich information from input images*, therefore we need a strong perturbation from random registers to mitigate the influence of already-captured patterns on attention maps.

An intuitive way to strengthen the perturbation is to add more random registers, making random registers inefficient. What if we add random registers to hinder the original attention maps $\sum_{k=1}^{n} e^{Q_i K_k^T}$ in Eq. 6? This would perturb the already captured information and intensify the perturbation that random registers bring. Based on this insight, we then design a kind of perturbation method on the input image tokens that contain rich semantic regions and contribute

most to attention maps, reflected in the attention maps as

$$A_{i,j} = \frac{e^{Q_i K_j^T}}{\underbrace{\sum_{k=1}^{m} e^{Q_i K_k^T}}_{\textit{Maintained Image}} + \underbrace{\sum_{k=1}^{n-m} e^{Q_i \overline{K}_k^T}}_{\textit{Image Perturbation}} + \underbrace{\sum_{k=1}^{\tilde{n}} e^{Q_i \tilde{K}_k^T}}_{\textit{Register Perturbation}}}, \quad (8)$$

where $m$ is the remaining number of original image patches, and $n-m$ is the number of perturbed image patches. Following (Naseer et al., 2021), ViT is highly robust to severe occlusions (e.g., random patch perturbations). Adding perturbations to random patches seems ill-advised. However, as the core mechanism of ViT, the attention map tends to rely on capturing patterns from continuous regions in the image (Wei et al., 2024). This inspires us to first cluster image tokens, then replace clusters with random registers to increase the perturbation to the attention maps.

Specifically, as shown in Fig. 6a, given an input image $I$ and its patches $X \in R^{n \times d}$ where $X_i$ is the average of pixels in the $i$th patch ($i \in n$), we randomly select a large portion of patches $a$ ($60\%n \leq a < n$) for every image as the anchors $A$ (e.g., the $j$th patch is selected as the anchor $A_j = X_j$, $j \in a$). Then, we compute the anchors' distance with image patches by leveraging the cosine similarity

$$cos(X_i, A_j) = \frac{X_i \cdot A_j}{||X_i|| \cdot ||A_j||}, \quad (9)$$

A cluster is defined as consisting of an anchor patch $A_j$ and corresponding patches within the similarity threshold $sim$

$$Cluster_{A_j} = \{X_i \mid cos(X_i, A_j) \geq sim, X_i \in X\}, \quad (10)$$

where the $sim$ is automatically computed to guarantee the drop ratio is stable. Then, we replace the clusters with random registers. The random registers are randomly sampled from Gaussian distribution $N(0,1) \cdot \tau$ where $\tau$ is a learnable parameter, written as

$$\tilde{T}_i = \begin{cases} X_i, X_i \notin Cluster \\ T^{R_i}, X_i \in Cluster \end{cases}, T^{R_i} \sim N(0, \tau^2) \quad (11)$$

We add additional random registers at the end of the input token sequence (Fig. 6), but the number of additionally added registers is small (verified in Fig. 10b). So the input of the model fed into the feature extractor is written as

$$f(P) = f(C(T^C, \tilde{T}_1, \ldots, \tilde{T}_n, T^{R_1}, \cdots, T^{R_{\tilde{n}}})), \quad (12)$$

By multiplied with randomized keys from random registers, the term $\sum_{k=1}^{n-m} e^{Q_i \times \tilde{K}_k^T}$ in Eq. 8 will also be the noise. So it can also be regarded as SAM.

### 3.2. Source-domain stage

In this stage, our goal is to help the model learn domain-agnostic information on the source domain. Based on the above analysis and interpretation. We utilize REAP to fully dig out the potential of random registers, helping the model learn more domain-agnostic information and find the flattened minimum in the loss landscape. Therefore, later the model can be well transferred to the target domain. The goal of our model in the source-domain stage is

$$L = \frac{1}{N} \sum_j^N L_{cls}(\phi(f(C(T^C, \tilde{T}, T^R), y_j^S)), \quad (13)$$

### 3.3. Target-domain stage

In this stage, our goal is to help the model adapt to target domains with only scarce data. Since the random register reduces the learning of domain information, it is not suitable at this stage. Therefore, we revisit the learnable register for its tendency to learn domain information, which will help the model in the few-shot adaptation to target domains. So we activate the additional registers to the learnable state (e.g., $T^L$), as shown in Fig. 6b, and finetune the model on the support set with a classifier for each episode as

$$L = \frac{1}{N} \sum_j^N L_{cls}(\phi(f(C(T^C, T(I), T^L), y_j^T)). \quad (14)$$

## 4. Experiments

### 4.1. Implementation Details

Following current works (Oh et al., 2022), our model is trained on the *mini*ImageNet dataset (Vinyals et al., 2016) as the source domain and then transferred to four target-domain datasets, including CropDiseases (Mohanty et al., 2016), EuroSAT (Helber et al., 2019), ISIC2018 (Codella et al., 2019), and ChestX (Wang et al., 2017), using the k-way n-shot classification.

During the training on the source domain, we take ViT-S as the backbone network and DINO pretraining on ImageNet as the initialization following (Zhang et al., 2022b; Fu et al., 2023). We set the ratio of anchor number and minimum drop ratio to 70% for cluster-dropping to the images, and replace them with Random Registers. We use the learnable standard deviation to generate random Gaussian noise and the initial value we set is 0.1. In addition, we also concatenate an additional 16 random registers to the reconstructed patches as the input sequence of the model. Our model has trained with the Adam (Kingma & Ba, 2017) optimizer for 50 epochs with a learning rate of $10^{-5}$ for the backbone network and $10^{-3}$ for the classifier respectively. During the few-shot evaluation on target domains, we provide the image with the same number(16) of learnable registers as the input of the ViT and set a learning rate of $10^{-3}$ for registers especially for absorbing target-domain domain-specific information.

*Table 1.* Comparison with state-of-the-art works based on ViT-S on target domains.

| Method | Shot | FT | Mark | ChestX | ISIC2018 | EuroSAT | CropDiseases | Average |
|---|---|---|---|---|---|---|---|---|
| MEM-FS (Walsh et al., 2023) | 1 | $\times$ | TIP-23 | 22.76 | 32.97 | 68.11 | 81.11 | 51.24 |
| StyleAdv (Fu et al., 2023) | 1 | $\times$ | CVPR-23 | 22.92 | 33.05 | 72.15 | 81.22 | 52.34 |
| FLoR (Zou et al., 2024a) | 1 | $\times$ | CVPR-24 | 22.78 | 34.20 | 72.39 | 81.81 | 52.80 |
| DAMIM (Ma et al., 2024) | 1 | $\times$ | AAAI-25 | 22.97 | 34.66 | 72.87 | 82.34 | 53.21 |
| CD-CLS (Zou et al., b) | 1 | $\times$ | NeurIPS-24 | 22.93 | 34.21 | 74.08 | 83.51 | 53.68 |
| AttnTemp (Zou et al., a) | 1 | $\times$ | NeurIPS-24 | 23.19 | 34.92 | 74.35 | 84.02 | 54.12 |
| **REAP** | 1 | $\times$ | **Ours** | **23.62** | **37.21** | **74.69** | **84.04** | **54.89** |
| PMF (Shell Xu, 2022) | 1 | $\checkmark$ | CVPR-22 | 21.73 | 30.36 | 70.74 | 80.79 | 50.91 |
| FLoR (Zou et al., 2024a) | 1 | $\checkmark$ | CVPR-24 | 23.26 | 35.49 | 73.09 | 83.55 | 53.85 |
| StyleAdv (Fu et al., 2023) | 1 | $\checkmark$ | CVPR-23 | 22.92 | 33.99 | 74.93 | 84.11 | 53.99 |
| DAMIM (Ma et al., 2024) | 1 | $\checkmark$ | AAAI-25 | 23.38 | 36.35 | 73.61 | 83.90 | 54.31 |
| CD-CLS (Zou et al., b) | 1 | $\checkmark$ | NeurIPS-24 | 23.39 | 35.56 | 74.97 | 84.54 | 54.62 |
| AttnTemp (Zou et al., a) | 1 | $\checkmark$ | NeurIPS-24 | 23.63 | 38.05 | 75.09 | 84.78 | 55.39 |
| **REAP** | 1 | $\checkmark$ | **Ours** | **24.17** | **38.67** | **75.97** | **85.33** | **56.04** |
| MEM-FS + RDA[*] (Walsh et al., 2023) | 1 | $\checkmark$ | TIP-23 | 23.85 | 37.07 | 75.91 | 83.74 | 55.14 |
| DAMIM[*] (Ma et al., 2024) | 1 | $\checkmark$ | AAAI-25 | 23.91 | 38.07 | 77.23 | 86.74 | 56.49 |
| CD-CLS (Zou et al., b) | 1 | $\checkmark$ | NeurIPS-24 | 23.88 | 37.20 | 78.41 | 87.39 | 56.72 |
| AttnTemp (Zou et al., a) | 1 | $\checkmark$ | NeurIPS-24 | 23.96 | **40.13** | 77.40 | 87.58 | 57.23 |
| **REAP[*]** | 1 | $\checkmark$ | **Ours** | **24.49** | 39.53 | **79.13** | **89.33** | **58.12** |
| MEM-FS (Walsh et al., 2023) | 5 | $\times$ | TIP-23 | 26.67 | 47.38 | 86.49 | 93.74 | 63.57 |
| StyleAdv (Fu et al., 2023) | 5 | $\times$ | CVPR-23 | 26.97 | 47.73 | 88.57 | 94.85 | 64.53 |
| FLoR (Zou et al., 2024a) | 5 | $\times$ | CVPR-24 | 26.71 | 49.52 | 90.41 | 95.28 | 65.48 |
| DAMIM (Ma et al., 2024) | 5 | $\times$ | AAAI-25 | 27.28 | 50.76 | 89.50 | 95.52 | 65.77 |
| CD-CLS (Zou et al., b) | 5 | $\times$ | NeurIPS-24 | 27.23 | 50.46 | **91.04** | 95.68 | 66.10 |
| AttnTemp (Zou et al., a) | 5 | $\times$ | NeurIPS-24 | 27.72 | **53.09** | 90.13 | 95.53 | 66.62 |
| **REAP** | 5 | $\times$ | **Ours** | **27.98** | 52.80 | 90.53 | **95.68** | **66.75** |
| PMF (Shell Xu, 2022) | 5 | $\checkmark$ | CVPR-22 | 27.27 | 50.12 | 85.98 | 92.96 | 64.08 |
| StyleAdv (Fu et al., 2023) | 5 | $\checkmark$ | CVPR-23 | 26.97 | 51.23 | 90.12 | 95.99 | 66.08 |
| FLoR (Zou et al., 2024a) | 5 | $\checkmark$ | CVPR-24 | 27.02 | 53.06 | 90.75 | 96.47 | 66.83 |
| DAMIM (Ma et al., 2024) | 5 | $\checkmark$ | AAAI-25 | 27.82 | 54.86 | 91.18 | 96.34 | 67.55 |
| CD-CLS (Zou et al., b) | 5 | $\checkmark$ | NeurIPS-24 | 27.66 | 54.69 | 91.53 | 96.27 | 67.54 |
| AttnTemp (Zou et al., a) | 5 | $\checkmark$ | NeurIPS-24 | 28.03 | 54.91 | 90.82 | 96.66 | 67.61 |
| **REAP** | 5 | $\checkmark$ | **Ours** | **28.34** | **55.28** | **91.79** | **96.71** | **68.03** |
| MEM-FS + RDA[*] (Walsh et al., 2023) | 5 | $\checkmark$ | TIP-23 | 27.98 | 51.02 | 88.77 | 95.04 | 65.70 |
| DAMIM[*] (Ma et al., 2024) | 5 | $\checkmark$ | AAAI-25 | 28.10 | 55.44 | 91.08 | 96.49 | 67.78 |
| CD-CLS[*] (Zou et al., b) | 5 | $\checkmark$ | NeurIPS-24 | 28.25 | 55.66 | 91.68 | 96.62 | 68.05 |
| AttnTemp[*] (Zou et al., a) | 5 | $\checkmark$ | NeurIPS-24 | 28.41 | 55.22 | 91.34 | 96.74 | 67.93 |
| **REAP[*]** | 5 | $\checkmark$ | **Ours** | **28.80** | **56.07** | **91.92** | **96.74** | **68.38** |

## 4.2. Comparison with State-of-the-Art Works

Comparisons with the state-of-the-art works are listed in Tab. 1, taking the ViT-S backbone for both 1-shot and 5-shot settings on four target-domain datasets. We group work by whether finetuning (FT) or utilizing the transductive (*) setting for fairness, following (Zou et al., a;b). As shown in Tab. 1, our works achieve the top average performance in all settings, demonstrating that the approach could consistently outperform current works. Please refer to the appendix for more comparisons (e.g., CNN-based SOTAs).

## 4.3. Ablation Study

The ablation study of each module is reported in Tab. 2. We can see both the random and image perturbation in

*Table 2.* Ablation study of source-domain training by 5-shot.

| Method | CropDisease | EuroSAT | ISIC2018 | ChestX | Ave. |
|---|---|---|---|---|---|
| Baseline | $94.61_{\pm0.17}$ | $89.29_{\pm0.17}$ | $46.16_{\pm0.23}$ | $26.21_{\pm0.17}$ | 64.07 |
| + Random Registers | $95.14_{\pm0.16}$ | $89.44_{\pm0.16}$ | $48.92_{\pm0.23}$ | $26.68_{\pm0.17}$ | 65.05 |
| **+ REAP** | $\mathbf{95.68}_{\pm0.15}$ | $\mathbf{90.53}_{\pm0.15}$ | $\mathbf{52.80}_{\pm0.24}$ | $\mathbf{27.98}_{\pm0.18}$ | **66.75** |
| (a) Random-mask | $91.23_{\pm0.19}$ | $84.42_{\pm0.22}$ | $43.89_{\pm0.22}$ | $24.06_{\pm0.16}$ | 60.90 |
| (b) Cluster-mask | $94.61_{\pm0.18}$ | $89.59_{\pm0.21}$ | $47.33_{\pm0.23}$ | $26.38_{\pm0.18}$ | 64.29 |

Eq. 8 contribute to the performance of target domains. We further make a comparison between our approaches with other similar works to validate the rationale of our designs.

### 4.3.1. IMAGE PERTURBATION BY CLUSTERING

To study the contribution of clustering, we randomly select the anchors in the image and mask these anchors in Tab. 2a.

*Table 3.* Ablation study of target-domain finetuning by 1-shot.

| Method | CropDisease | EuroSAT | ISIC2018 | ChestX | Ave. |
|---|---|---|---|---|---|
| Regular Finetuning | $83.61_{\pm0.19}$ | $74.50_{\pm0.21}$ | $36.52_{\pm0.25}$ | $23.36_{\pm0.20}$ | 54.50 |
| Random Registers | $83.13_{\pm0.17}$ | $74.01_{\pm0.24}$ | $35.53_{\pm0.21}$ | $23.34_{\pm0.17}$ | 54.00 |
| **Learnable Registers** | $\mathbf{85.33_{\pm0.17}}$ | $\mathbf{75.97_{\pm0.25}}$ | $\mathbf{38.67_{\pm0.19}}$ | $\mathbf{24.17_{\pm0.18}}$ | **56.04** |

For a fair comparison, we keep the same masking ratio in both clustering and random approaches. We can see the performance is even much lower than the baseline, which means the random type would harm the model, and verifying the image perturbation by clusters is crucial.

#### 4.3.2. REPLACING WITH RANDOM REGISTERS

We study the contribution of replacing the clusters with random registers. We directly remove the clusters (Tab. 2b). The performance is lower than ours, indicating that replacing the clusters with random registers is an effective way to help the model be more transferable.

#### 4.3.3. TARGET-DOMAIN FINETUNING

To study the contribution to target-domain finetuning, we compare the regular finetuning operation, finetuning with random registers, and finetuning with learnable registers. As shown in Tab. 3, finetuning with learnable registers could improve the model's performance on the target domains while random registers decrease it, which further verifies the absorption and perturbation of domain information in learnable and random registers, respectively.

### 4.4. Verification of model generalization

#### 4.4.1. QUANTITATIVE STUDY

As shown in Fig. 7, we calculate the domain similarity of the features extracted from the trained backbone network between source and target domains by the CKA similarity. We can see our model significantly increases the domain similarity, indicating our method encourages the model to learn domain-agnostic information of well transferability.

#### 4.4.2. QUALITATIVE STUDY

The visualization of the attention maps in our model on both the source and target domain is shown in Fig. 8. Compared to the dispersed attention observed in the baseline, our model focuses on more valid and concentrated regions within the image, verifying that our approach improves the generalization to target domains.

### 4.5. Sensitivity Study of Hyper-parameters

We study the hyper-parameters in Fig. 9, 10, 11 and see that:

(1) The input layer is effective. In Fig. 9a, only applying our approach in the first block (input layer) could significantly improve the model's performance on target domains,

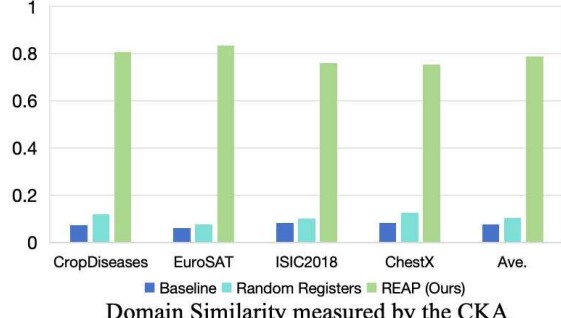

Domain Similarity measured by the CKA

*Figure 7.* Utilizing our approaches significantly increases the domain similarity, proving that utilizing our approach makes the model more domain-agnostic.

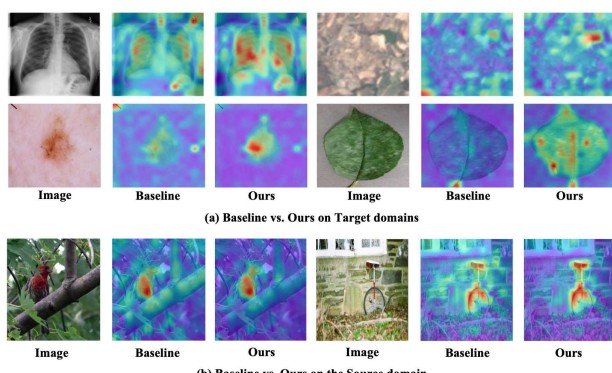

*Figure 8.* (a) The heatmap on target domains proves that our approaches effectively improve target-domain performance. (b) The heatmap on the source domain demonstrates that our approach effectively avoids overfitting on the source domain.

proving that the information in the image is the most representable and influence the whole attention maps than others.

(2) A high anchor ratio is better, but too high harm. From Fig. 9b, the ratio of anchor number sets between 40% and 80% significantly improves performance. A rising anchor number means that more continual areas tend to be replaced, but too much will destroy important information.

(3) A high replaced ratio is better, but cannot be too high. As we can see in Fig. 10a, setting the replaced ratio higher could see a steady rise in the performance on target domains before 70%. After that, the performance drops sharply. The same situation happens with the anchor number ratio, indicating that a higher replaced ratio could help perturb more on the attention maps, while a ratio that is too high may likely throw away too much information for learning.

(4) Additional random registers are beneficial. In Fig. 10b, after replacing the clusters with random registers, concatenating some random registers further improves the performance on the target domain. Setting the extra register number to 16 is better than less and more. Notably, this number of registers is much smaller than that in Fig. 1b.

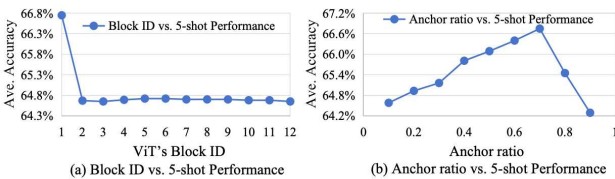

*Figure 9.* (a) Applying our approach only on the first block (input layer) can improve the performance. (b) A high anchor ratio can effectively improve performance.

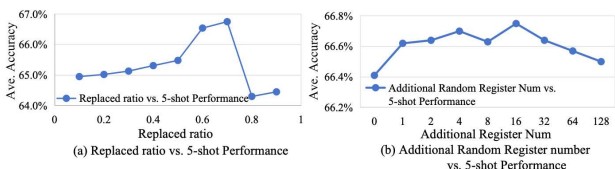

*Figure 10.* (a) A high replaced ratio can effectively improve performance, but too high would harm. (b) Additional random registers are beneficial, and the number is much smaller than that in Fig. 1b.

(5) Moderate perturbation helps. In Fig. 11, when we set a consistent number of random registers (e.g., 16) and gradually increase the standard deviation of their normal distribution, the model's performance initially rises and then declines, attesting to the necessity of a moderate level of perturbation to the attention map, neither too slight nor excessive.

### 4.6. Applying Our Method to Other Backbones

We also implement our approach on different backbones, like ViT pretrained by iBOT(Zhou et al., 2021), ViT-Base(Zhang et al., 2022b) pretrained by DINO, and ViT-Base pretrained by CLIP(Radford et al., 2021). The results can be seen in Tab. 4. Specifically, iBOT represents the iBOT-pretrained ViT baseline, DINO-ViT-Base corresponds to the ViT-Base pretrained by DINO baseline and CLIP corresponds to the ViT-Base pretrained by CLIP baseline. It is clarified from the average performance of four target domains that our approach shows considerable improvement among all backbones in a 5-way 5-shot setting.

## 5. Related Work

**Cross-Domain Few-Shot Learning (CDFSL)** was proposed in FWT (Tseng et al., 2020) and has a new benchmark in BSCD-FSL (Guo et al., 2020). Recently, it has been studied by several works (Oh et al., 2022; Zhang et al., 2022c; Xu et al., 2023; 2024), which focuses on training a model on the source domain that can generalize well to target domain with limited examples. Current works can be grouped into two types: meta-learning based approaches (Fu et al., 2022; Hu & Ma, 2022), learning task-agnostic knowledge to learn new tasks efficiently (Guo et al., 2020), and transfer learning based approaches (Zhou et al., 2023; Zou et al., 2024a), reusing the model trained on the base classes dataset. However, there has been insufficient in-depth research into the

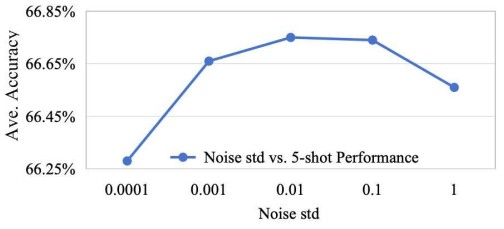

*Figure 11.* Setting the random registers with a standard deviation of intermediate magnitude is optimal, demonstrating that the disturbance to attention maps should neither be too weak nor too strong.

*Table 4.* Ablation study of our method with more backbones.

| Method | CropDiseases | EuroSAT | ISIC2018 | ChestX | Ave. |
|---|---|---|---|---|---|
| CLIP | $93.08_{\pm0.18}$ | $74.34_{\pm0.25}$ | $41.29_{\pm0.23}$ | $\mathbf{23.98}_{\pm0.16}$ | 58.17 |
| **CLIP + Ours** | $\mathbf{93.93}_{\pm0.16}$ | $\mathbf{81.10}_{\pm0.23}$ | $\mathbf{44.90}_{\pm0.24}$ | $23.78_{\pm0.16}$ | **60.93** |
| iBOT | $94.01_{\pm0.17}$ | $88.80_{\pm0.16}$ | $43.88_{\pm0.24}$ | $25.32_{\pm0.16}$ | 63.00 |
| **iBOT + Ours** | $\mathbf{94.88}_{\pm0.17}$ | $\mathbf{89.18}_{\pm0.17}$ | $\mathbf{46.71}_{\pm0.23}$ | $\mathbf{26.36}_{\pm0.17}$ | **64.28** |
| DINO-ViT-Base | $95.39_{\pm0.15}$ | $89.29_{\pm0.16}$ | $47.98_{\pm0.24}$ | $26.29_{\pm0.17}$ | 64.74 |
| **DINO-ViT-Base + Ours** | $\mathbf{96.11}_{\pm0.14}$ | $\mathbf{89.77}_{\pm0.16}$ | $\mathbf{50.88}_{\pm0.24}$ | $\mathbf{26.70}_{\pm0.17}$ | **65.87** |

performance of ViT in extreme cross-domain scenarios.

**Prompts** are widely used in natural language processing (Yao et al., 2023), which refers to a given one or series of text inputs intended to guide or trigger the model to produce a specific output or response (Liu et al., 2023). As large language models progress, Prompt has played an important role (White et al., 2023). In computer vision, ViT has been widely used as a backbone. The CLS token of ViT (Liu et al., 2021) is also a special kind of prompt of vital importance for the classification (Zhang et al., 2022a). Recently, prompt-based approaches (Yao et al., 2023; Wang et al., 2023; Sohn et al., 2023) are developed for various downstream vision tasks, like (Darcet et al., 2024) extend the input sequence with prompts without adding any information and drop before the output, successfully avoiding collateral side effects. ProD (Ma et al., 2023) uses two parallel prompts to disentangle the domain-general and domain-specific knowledge to alleviate the domain gap. However, it merely utilizes prompts without considering or explaining the overfitting problem in prompt tuning within the source domain. In all, prompt learning under large domain gaps is still under-explored.

## 6. Conclusion

In this paper, we find a phenomenon that providing additional learning registers is detrimental to the CDFSL performance while adding random noise to registers improves it. We delve into this phenomenon for an interpretation and find that the learnable registers naturally absorb domain information, while random registers perturb it and help the model find a flattened minimum in the loss landscapes of well transferability. Based on these, we further propose a method to make random registers more effective and efficient. Experiments validate our rationale and effectiveness.

## Acknowledgments

This work is supported by the National Key Research and Development Program of China under grant 2024YFC3307900; the National Natural Science Foundation of China under grants 62206102, 62436003, 62376103 and 62302184; Major Science and Technology Project of Hubei Province under grant 2024BAA008; Hubei Science and Technology Talent Service Project under grant 2024DJC078; and Ant Group through CCF-Ant Research Fund. The computation is completed in the HPC Platform of Huazhong University of Science and Technology.

## Impact Statement

We introduce a CD-FSL method that enhances perturbations in attention maps by adding random registers to the semantic regions of image tokens in the ViT. This helps mitigate the domain gap and improves generalization to the target domain. Our approach is also applicable to other areas, such as domain generalization, domain adaptation, and few-shot class-incremental learning, where improving model transferability is a common challenge. While our evaluations focus on four distinct target domains, these may not encompass all potential real-world scenarios. Therefore, further evaluation across a wider range of target domains is needed to validate the approach in more realistic settings.

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

# Appendix for Random Registers for Cross-Domain Few-Shot Learning

## A. Detailed Dataset Description

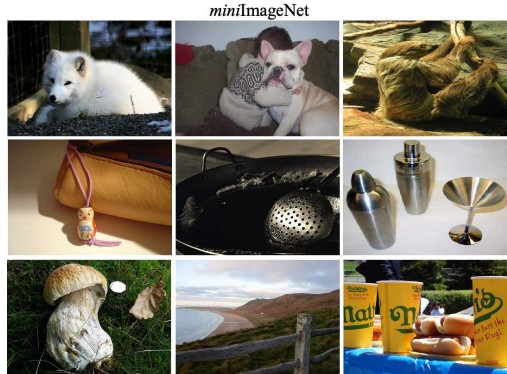

*Figure 12.* Samples of the source-domain *mini*ImageNet dataset.

***mini*ImageNet** (Vinyals et al., 2016) is a widely used dataset in the field of meta-learning and few-shot learning. It is derived from ImageNet (Deng et al., 2009) dataset but with a smaller scale. The *mini*ImageNet dataset contains a total of 60,000 natural images in 100 categories, each with 600 samples and each image size 84 x 84 pixels. As shown in Fig. 12, the images in the *mini*ImageNet dataset come from various scenes, including but not limited to social scenes, some of which include human objects, while others are photos taken in real scenes, displaying diverse content and features. Following the current works (Oh et al., 2022; Tseng et al., 2020), we split it into 64 base classes as the source-domain dataset for training. Additionally, as depicted in Fig. 13, we utilize datasets from four distinct domains as target-domain datasets, including plant disease, surface satellite imagery, skin disease, and chest X-ray images respectively. We'll introduce them sequentially below.

**CropDiseases** (Mohanty et al., 2016) is an essential resource for agricultural disease identification. It is composed of high-resolution and highly similar original images of similar crop diseases. Specifically, it consists of 38 distinct classes and a total of 43,456 images, which are natural images but are very specialized, including various infected crops, healthy plants, and their corresponding disease category labels.

**EuroSAT** (Helber et al., 2019) is a remote sensing image dataset for land use and cover classification, containing a total of 27,000 satellite images of the Earth categorized into 10 distinct classes. The images in the EuroSAT are less similar to images in *mini*ImageNet since they lack perspective distortion, but still color images of natural scenes.

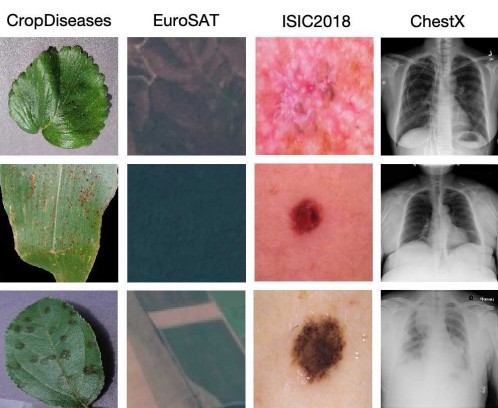

*Figure 13.* Samples of target-domain datasets: CropDiseases, EuroSAT, ISIC2018, and ChestX.

**ISIC2018** (Codella et al., 2019) is an important skin disease image dataset for the classification of dermoscopic images, encompassing 10,015 medical images for skin lesion classification across 7 different classes. It is even less similar to the *mini*ImageNet as it could not even represent natural scenes.

**ChestX** (Wang et al., 2017) is a medical imaging dataset containing 25,847 frontal X-ray images distributed across 7 distinct classes. The dataset is the most dissimilar to the *mini*ImageNet in three orthogonal criteria. Apart from the two factors mentioned above, losing 2 color channels appears in the ChestX.

## B. Detailed Descriptions of the CKA

Following current works (e.g., (Oh et al., 2022)), the domain similarity is measured by comparing the distance between two batches of images, where each batch is sampled from a single domain. We follow (Kornblith et al., 2019) to take it as the similarity function.

The CKA is the abbreviation of the Centered Kernel Alignment, measuring the similarity between feature representations. Based on the idea of kernel function, CKA measures the similarity between samples in two feature spaces by calculating their kernel matrix. However, unlike traditional kernel functions, CKA also introduces centralized operations to ensure that the contribution of each sample is taken into account when aligned. The detailed process of CKA (Central-core Alignment) analysis consists of the following steps and formulas:

*Table 5.* Comparison with more state-of-the-art works based by 5-way 1-shot accuracy.

| Method | Backbone | FT | Mark | ChestX | ISIC2018 | EuroSAT | CropDiseases | Average |
|---|---|---|---|---|---|---|---|---|
| GNN + FT (Tseng et al., 2020) | ResNet10 | × | ICLR-20 | 22.00 | 30.22 | 55.53 | 60.74 | 42.12 |
| MN + AFA (Hu & Ma, 2022) | ResNet10 | × | ECCV-22 | 22.11 | 32.32 | 61.28 | 60.71 | 44.10 |
| GNN + ATA (Wang & Deng, 2021) | ResNet10 | × | IJCAI-21 | 22.10 | 33.21 | 61.35 | 67.47 | 46.53 |
| GNN + AFA (Hu & Ma, 2022) | ResNet10 | × | ECCV-22 | 22.92 | 33.21 | 63.12 | 67.61 | 46.97 |
| LDP-net (Zhou et al., 2023) | ResNet10 | × | CVPR-23 | 23.01 | 33.97 | 65.11 | 69.64 | 47.18 |
| FLoR (Zou et al., 2024a) | ResNet10 | × | CVPR-24 | 23.11 | **38.11** | 62.90 | 73.64 | 49.69 |
| MEM-FS (Walsh et al., 2023) | ViT-S | × | TIP-23 | 22.76 | 32.97 | 68.11 | 81.11 | 51.24 |
| StyleAdv (Fu et al., 2023) | ViT-S | × | CVPR-23 | 22.92 | 33.05 | 72.15 | 81.22 | 52.34 |
| FLoR (Zou et al., 2024a) | ViT-S | × | CVPR-24 | 22.78 | 34.20 | 72.39 | 81.81 | 52.80 |
| DAMIM (Ma et al., 2024) | ViT-S | × | AAAI-25 | 22.97 | 34.66 | 72.87 | 82.34 | 53.21 |
| CD-CLS (Zou et al., b) | ViT-S | × | NeurIPS-24 | 22.93 | 34.21 | 74.08 | 83.51 | 53.68 |
| AttnTemp (Zou et al., a) | ViT-S | × | NeurIPS-24 | 23.19 | 34.92 | 74.35 | 84.02 | 54.12 |
| **REAP** | ViT-S | × | **Ours** | **23.62** | 37.21 | **74.69** | **84.04** | **54.89** |
| PMF (Shell Xu, 2022) | ViT-S | ✓ | CVPR-22 | 21.73 | 30.36 | 70.74 | 80.79 | 50.91 |
| FLoR (Zou et al., 2024a) | ViT-S | ✓ | CVPR-24 | 23.26 | 35.49 | 73.09 | 83.55 | 53.85 |
| StyleAdv (Fu et al., 2023) | ViT-S | ✓ | CVPR-23 | 22.92 | 33.99 | 74.93 | 84.11 | 53.99 |
| DAMIM (Ma et al., 2024) | ViT-S | ✓ | AAAI-25 | 23.38 | 36.35 | 73.61 | 83.90 | 54.31 |
| CD-CLS (Zou et al., b) | ViT-S | ✓ | NeurIPS-24 | 23.39 | 35.56 | 74.97 | 84.54 | 54.62 |
| AttnTemp (Zou et al., a) | ViT-S | ✓ | NeurIPS-24 | 23.63 | 38.05 | 75.09 | 84.78 | 55.39 |
| **REAP** | ViT-S | ✓ | **Ours** | **24.17** | **38.67** | **75.97** | **85.33** | **56.04** |
| LDP-net[*] (Zhou et al., 2023) | ResNet10 | ✓ | CVPR-23 | 22.21 | 33.44 | 73.25 | 81.24 | 52.54 |
| TPN + ATA[*] (Wang & Deng, 2021) | ResNet10 | ✓ | IJCAI-21 | 22.45 | 35.55 | 70.84 | 82.47 | 52.83 |
| RDC[*] (Li et al., 2022) | ResNet10 | ✓ | CVPR-22 | 22.32 | 36.28 | 70.51 | 85.79 | 53.73 |
| MEM-FS + RDA[*] (Walsh et al., 2023) | ViT-S | ✓ | TIP-23 | 23.85 | 37.07 | 75.91 | 83.74 | 55.14 |
| DAMIM[*] (Ma et al., 2024) | ViT-S | ✓ | AAAI-25 | 23.91 | 38.07 | 77.23 | 86.74 | 56.49 |
| CD-CLS (Zou et al., b) | ViT-S | ✓ | NeurIPS-24 | 23.88 | 37.20 | 78.41 | 87.39 | 56.72 |
| AttnTemp (Zou et al., a) | ViT-S | ✓ | NeurIPS-24 | 23.96 | **40.13** | 77.40 | 87.58 | 57.23 |
| **REAP[*]** | ViT-S | ✓ | **Ours** | **24.49** | 39.53 | **79.13** | **89.33** | **58.12** |

Create a Gram matrix: First, for the given two sets of features representing $X$ and $Y$, compute their Gram matrices $K$ and $L$. These Gram matrices are obtained by calculating the inner product between feature representations, i.e. $K = X \cdot X^T$ and $L = Y \cdot Y^T$. Centralized Gram matrix: Next, the Gram matrices K and L are centralized to eliminate the impact of the average value of the data on the results. Calculate Hilbert-Schmidt independence criterion (HSIC): Use vectorization operation ($vec(\cdot)$) to process the centralized Gram matrix, and then calculate the HSIC value. HSIC is calculated as follows

$$HSIC(K, L) = \frac{vec(K) \cdot vec(L)}{(N-1)^2}, \quad (15)$$

Where $vec(\cdot)$ represents the vectorization operation and N is the number of input samples. Final calculation of CKA: Finally, CKA calculates the similarity between two sets of feature representations by standardizing the HSIC. The calculation formula of CKA is as follows

$$\text{CKA}(XY) = \frac{HSIC(XX^T, YY^T)}{\sqrt{HSIC(XX^T, XX^T) HSIC(YY^T, YY^T)}}, \quad (16)$$

CKA is a standardized metric that represents the similarity of two Gram matrices $K$ and $L$. Since the Gram matrix reflects the feature relationship between the sample pairs,

CKA can be interpreted as the similarity of the relationship between the features in X and Y from different domains.

Therefore, we quantitatively measure the domain distance between source and target datasets by the CKA similarity following (Davari et al., 2022). Specifically, given a backbone network, we extract features from images in different domains and then calculate the CKA similarity by aligning the channel dimension. Suppose a model is completely overfitted to one domain, given other domains' images, the extracted features could be just random noises, therefore the domain similarity will be downgraded to 0. Therefore, following current works (e.g., (Kim & Han, 2023)), we hold that the larger domain similarity indicates the less domain information.

## C. Sharpness-Aware Minimization

Due to the domain shift between source and target domains, the training loss value on source-domain datasets offers limited assurances regarding the model's generalization ability. Therefore, merely optimizing the training loss value can steadily result in suboptimal model quality. Theoretically, from the sharpness of the loss landscapes, the sharper the minimum is the more vulnerable against the domain gaps

*Table 6.* Comparison with more state-of-the-art works by 5-way 5-shot accuracy.

| Method | Backbone | FT | Mark | ChestX | ISIC2018 | EuroSAT | CropDiseases | Average |
|---|---|---|---|---|---|---|---|---|
| MN + AFA (Hu & Ma, 2022) | ResNet10 | × | ECCV-22 | 23.18 | 39.88 | 69.63 | 80.07 | 53.19 |
| GNN + FT (Tseng et al., 2020) | ResNet10 | × | ICLR-20 | 24.28 | 40.87 | 78.02 | 87.07 | 57.06 |
| GNN + ATA (Wang & Deng, 2021) | ResNet10 | × | IJCAI-21 | 24.32 | 44.91 | 83.75 | 90.59 | 60.39 |
| LDP-net (Zhou et al., 2023) | ResNet10 | × | CVPR-23 | 26.67 | 48.06 | 82.01 | 89.40 | 61.29 |
| GNN + AFA (Hu & Ma, 2022) | ResNet10 | × | ECCV-22 | 25.02 | 46.01 | 85.58 | 88.06 | 61.67 |
| FLoR (Zou et al., 2024a) | ResNet10 | × | CVPR-24 | 26.70 | 51.44 | 80.87 | 91.25 | 62.32 |
| MEM-FS (Walsh et al., 2023) | ViT-S | × | TIP-23 | 26.67 | 47.38 | 86.49 | 93.74 | 63.57 |
| StyleAdv (Fu et al., 2023) | ViT-S | × | CVPR-23 | 26.97 | 47.73 | 88.57 | 94.85 | 64.53 |
| FLoR (Zou et al., 2024a) | ViT-S | × | CVPR-24 | 26.71 | 49.52 | 90.41 | 95.28 | 65.48 |
| DAMIM (Ma et al., 2024) | ViT-S | × | AAAI-25 | 27.28 | 50.76 | 89.50 | 95.52 | 65.77 |
| CD-CLS (Zou et al., b) | ViT-S | × | NeurIPS-24 | 27.23 | 50.46 | **91.04** | 95.68 | 66.10 |
| AttnTemp (Zou et al., a) | ViT-S | × | NeurIPS-24 | 27.72 | **53.09** | 90.13 | 95.53 | 66.62 |
| **REAP** | ViT-S | × | **Ours** | **27.98** | 52.80 | 90.53 | **95.68** | **66.75** |
| PMF (Shell Xu, 2022) | ViT-S | ✓ | CVPR-22 | 27.27 | 50.12 | 85.98 | 92.96 | 64.08 |
| StyleAdv (Fu et al., 2023) | ViT-S | ✓ | CVPR-23 | 26.97 | 51.23 | 90.12 | 95.99 | 66.08 |
| FLoR (Zou et al., 2024a) | ViT-S | ✓ | CVPR-24 | 27.02 | 53.06 | 90.75 | 96.47 | 66.83 |
| DAMIM (Ma et al., 2024) | ViT-S | ✓ | AAAI-25 | 27.82 | 54.86 | 91.18 | 96.34 | 67.55 |
| CD-CLS (Zou et al., b) | ViT-S | ✓ | NeurIPS-24 | 27.66 | 54.69 | 91.53 | 96.27 | 67.54 |
| AttnTemp (Zou et al., a) | ViT-S | ✓ | NeurIPS-24 | 28.03 | 54.91 | 90.82 | 96.66 | 67.61 |
| **REAP** | ViT-S | ✓ | **Ours** | **28.34** | **55.28** | **91.79** | **96.71** | **68.03** |
| ConFeSS* (Das et al., 2022) | ResNet10 | ✓ | ICLR-2022 | 27.09 | 48.85 | 84.65 | 88.88 | 62.37 |
| LDP-net* (Zhou et al., 2023) | ResNet10 | ✓ | CVPR-23 | 26.88 | 48.44 | 84.05 | 91.89 | 62.82 |
| RDC* (Li et al., 2022) | ResNet10 | ✓ | CVPR-22 | 25.07 | 49.91 | 84.29 | 93.30 | 63.14 |
| TPN + ATA* (Wang & Deng, 2021) | ResNet10 | ✓ | IJCAI-21 | 24.74 | 49.83 | 85.47 | 93.56 | 63.40 |
| MEM-FS + RDA* (Walsh et al., 2023) | ViT-S | ✓ | TIP-23 | 27.98 | 51.02 | 88.77 | 95.04 | 65.70 |
| DAMIM* (Ma et al., 2024) | ViT-S | ✓ | AAAI-25 | 28.10 | 55.44 | 91.08 | 96.49 | 67.78 |
| CD-CLS* (Zou et al., b) | ViT-S | ✓ | NeurIPS-24 | 28.25 | 55.66 | 91.68 | 96.62 | 68.05 |
| AttnTemp* (Zou et al., a) | ViT-S | ✓ | NeurIPS-24 | 28.41 | 55.22 | 91.34 | 96.74 | 67.93 |
| **REAP*** | ViT-S | ✓ | **Ours** | **28.80** | **56.07** | **91.92** | **96.74** | **68.38** |

*Table 7.* Comparison study of Our method with the baseline, image-perturbation, feature perturbation, weight perturbation, and attention maps direct perturbation by the 5-way 5-shot accuracy.

| Method | CropDiseases | EuroSAT | ISIC2018 | ChestX | Ave. |
|---|---|---|---|---|---|
| Baseline | $94.61_{\pm0.17}$ | $89.29_{\pm0.17}$ | $46.16_{\pm0.23}$ | $26.21_{\pm0.17}$ | 64.07 |
| $img_p$ | $95.14_{\pm0.15}$ | $89.73_{\pm0.17}$ | $46.30_{\pm0.23}$ | $26.54_{\pm0.17}$ | 64.43 |
| $fea_p$ | $94.78_{\pm0.16}$ | $88.76_{\pm0.17}$ | $47.68_{\pm0.23}$ | $26.80_{\pm0.17}$ | 64.51 |
| $weight_p$ | $95.12_{\pm0.16}$ | $89.73_{\pm0.17}$ | $46.42_{\pm0.23}$ | $26.36_{\pm0.17}$ | 64.41 |
| $attn_p$ | $94.90_{\pm0.16}$ | $89.03_{\pm0.17}$ | $46.09_{\pm0.23}$ | $25.90_{\pm0.17}$ | 63.98 |
| **Ours** | $95.68_{\pm0.15}$ | $90.53_{\pm0.15}$ | $52.80_{\pm0.24}$ | $27.98_{\pm0.18}$ | **66.75** |

model will be. Specifically, the sharpness is measured as

$$Sharpness = max_{\|\epsilon\|_2 \leq \rho} [L(w + \epsilon) - L(w)], \quad (17)$$

where $w$ refers to the model weights, $\epsilon$ refers to the perturbation with the radius $\rho$. With this criterion, the generalization can be bounded as follows

$$L_{\mathcal{D}}(w) \leq max_{\|\epsilon\|_2 \leq \rho} L_S((w + \epsilon)$$
$$+ \sqrt{\frac{k \log(1 + \frac{\|w\|_2^2}{\rho^2}(1 + \sqrt{\frac{log(n)}{k}})^2) + 4\log \frac{n}{\delta} + \tilde{O}(1)}{n - 1}}, \quad (18)$$

For any $\rho > 0$ and any distribution $\mathcal{D}$, with probability $1 - \delta$ over the choice of the training set $S \sim \delta$, where

$n = |S|$, k is the number of parameters and we assumed $L_{\mathcal{D}}(w) < \mathbb{E}_{\epsilon \sim (0,\rho)}[L_{\mathcal{D}}(w + \epsilon)]$. Following this criterion, we measure the sharpness given perturbations on different model weights to compare the optimal loss achieved by those methods. In the meantime, Sharpness Aware Minimization (SAM (Foret et al., 2021)) is introduced to improve the model's generalization ability, simultaneously minimizing loss value and loss sharpness

$$L_{SAM} = \underbrace{max_{\|\epsilon\|_2 \leq \rho} [L(w + \epsilon) - L(w)]}_{Sharpness} + \underbrace{L(w)}_{Loss\ Value} \quad (19)$$

The term enclosed in square brackets quantifies the sharpness by assessing how rapidly the training loss escalates when transitioning from w to a neighboring parameter value. The whole term aims at seeking parameters that reside in neighborhoods with uniformly low loss values rather than solely focusing on parameters with low loss values themselves, written as

$$min_w L_{SAM} = min_w [max_{\|\epsilon\|_2 \leq \rho} L(w + \epsilon)] \quad (20)$$

To minimize $L_{SAM}$, various perturbation approaches have been proposed and divided into three types: image perturbation (Foret et al., 2021), feature perturbation (Li et al., 2020), and weight perturbation (Liang et al., 2022). Image perturbation works by adding perturbation to the image ignoring

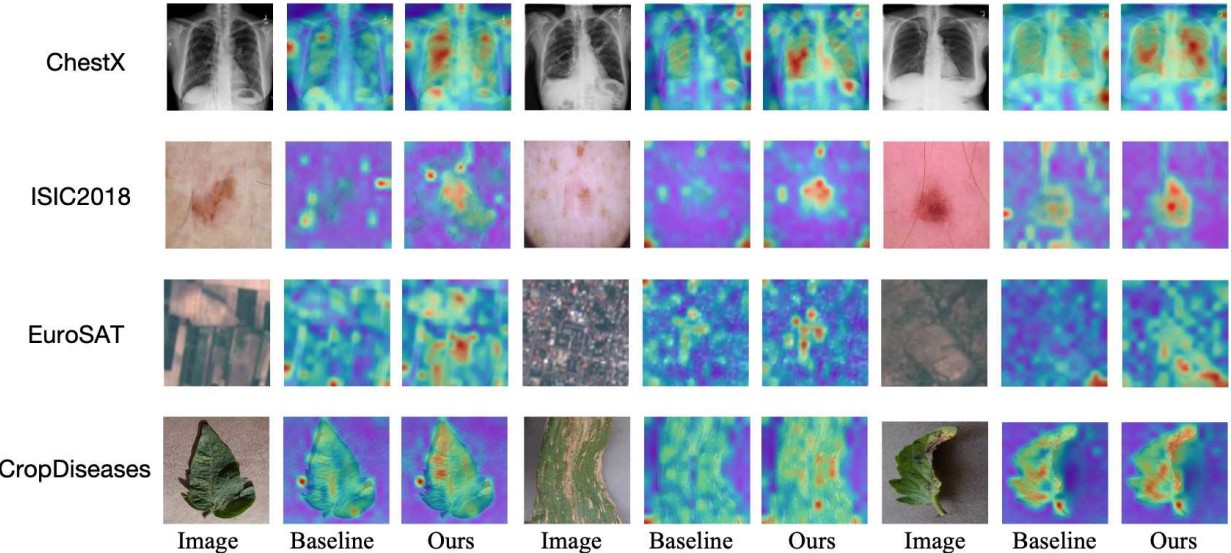

*Figure 14.* The heatmap on target domains proves that our approaches effectively improve the model's target-domain performance.

the model's structure. Feature perturbation is not limited to the first layer, but can be flexibly used in every layer of the model, while weight perturbation is directly adding perturbations to the model parameters. We compare our methods with these three classical methods in Tab. 7. Due to ignoring the idiosyncrasies of ViT, which is highly robust to severe occlusions (Naseer et al., 2021) (e.g., random patch perturbations), all these perturbations are a little useful but not effective enough when compared with the baseline. The core structure of the ViT is the attention mechanism, which plays an essential role in the whole architecture (Chen et al., 2023a). Instead of perturbing the whole weight parameters, mainly focusing on the attention maps may be a good idea. However, directly adding perturbations to the attention maps harms the performance. Our method, working as a new perturbation to attention maps indirectly, is proved to alleviate the domain shift problem in Tab. 7.

## D. More Experiments

### D.1. Comparison with more SOTAs

As illustrated in Tab. 5 and Tab. 6, we conduct a thorough comparison of various ViT and CNN-based approaches on CDFSL tasks. Our proposed methods consistently surpass all other approaches, achieving optimal performance. These findings emphasize the efficacy of our approach.

### D.2. More Visualization of the attention maps

The visualization of the attention maps in our model on the target domain is shown in Fig. 14. Compared to the dispersed attention observed in the baseline, our model focuses on more valid and concentrated regions within the image, verifying that our approach improves the generaliza-

tion from source domains to target domains.

### D.3. Applying the Learnable Registers to different layers of the model

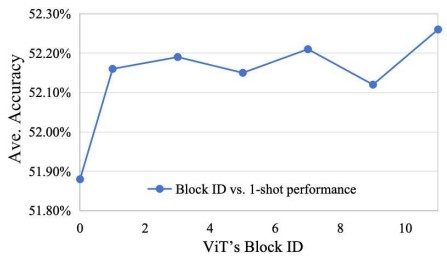

Block ID of Adding Learnable Registers vs. 1-shot performance

*Figure 15.* The model applied the learnable registers in the first block have the poorest performance on target domains, validating that due to the registers' location, the model tends to view the registers in the shallow layer (especially the input layer) as a kind of domain-relevant pattern.

In Fig. 15, when we keep the registers number equal to 4, the model applied the learnable registers in the first block (input layer) performs poorest on target domains, proving that due to the registers' location, the model tends to view the registers in the shallow layer (especially the input layer) as a kind of domain-relevant pattern.

### D.4. Similar Impact on deep prompts

As shown in the Fig. 16, the model with deep prompts shows similar trends towards sharpness with the shallow types, quantitatively verifying that random registers improve the transferability of attention in both deep types or shallow types.

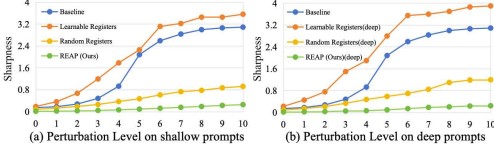

*Figure 16.* The model that applies deep registers has a similar impact on sharpness.

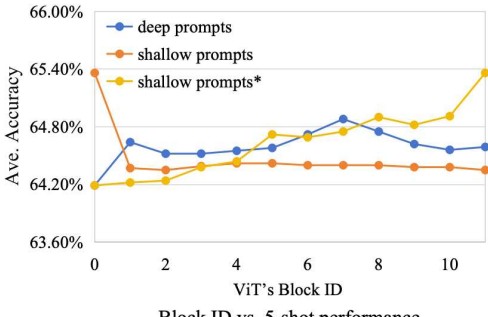

Block ID vs. 5-shot performance

*Figure 17.* The model applies shallow random registers from the input layer and retains them until the final layer has the best performance on target domains.

As depicted in the Fig. 17, we uniformly set the number of random registers to 1024 and add them at the end of the token sequence using both deep and shallow approaches before feeding them into the ViT blocks. The shallow registers without an asterisk (*) denote adding random registers starting from the $nth$ layer, while those with an asterisk (*) indicate removal after the nth layer. Among these, the approach of adding shallow random registers from the input layer and retaining them until the final layer yields the optimal performance.

*Table 8.* Validations of REAP in both deep and shallow prompts of source-domain training by 5-shot.

| Method | CropDisease | EuroSAT | ISIC2018 | ChestX | Ave. |
|---|---|---|---|---|---|
| Baseline | $94.61_{\pm0.17}$ | $89.29_{\pm0.17}$ | $46.16_{\pm0.23}$ | $26.21_{\pm0.17}$ | 64.07 |
| REAP w/ deep random prompts | $95.04_{\pm0.14}$ | $88.05_{\pm0.18}$ | $\mathbf{53.40}_{\pm0.23}$ | $27.83_{\pm0.18}$ | 66.08 |
| **REAP w/ shallow random prompts** | $\mathbf{95.68}_{\pm0.15}$ | $\mathbf{90.53}_{\pm0.15}$ | $52.80_{\pm0.24}$ | $\mathbf{27.98}_{\pm0.18}$ | **66.75** |

As illustrated in Tab. 8, our method effectively enhances performance regardless of whether prompts are utilized in deep or shallow mode.

