# OpenReview forum: "Random Registers for Cross-Domain Few-Shot Learning"
_ICML.cc/2025/Conference — ICML 2025 poster_

### Official Review · Reviewer_cSmv · 2025-03-11

**Overall Recommendation:** 3

**Summary:**

In this work, the authors propose a two-stage learning framework namely REAP to tackle CDFSL problem. During the source domain training, REAP randomly masks the most discriminative region and fill the erased region by random prompts, then optimize the pretrained ViT on source data. And during target domain fine-tuning, REAP optimizes the learnable prompt to adapt to target data. The proposed REAP achieves comparable performance among all methods on multiple benchmarks.

**Claims And Evidence:**

The claim is relative weak, previous work [1] manages to discuss some similar experimental results regarding learning prompts on source dataset and direct evaluation on target datasets, but prompt tuning is enough to tackle this issue.

[1] Learning to Prompt for Vision-Language Models, IJCV

**Essential References Not Discussed:**

[1] Visual Prompt Tuning, ECCV 2022

**Experimental Designs Or Analyses:**

The experimental results are not sufficient, some important analysis is lacked. For example, the ablation study of random erasing strategy (i.e., simply masking but not using random registers to fill), initialization methods of random registry, and comparison between the proposed method and VPT.

**Methods And Evaluation Criteria:**

Though the authors discuss the problem from sharpness aware minimization aspect, the proposed method is still a vanilla pretraining on source and prompt tuning on target paradigm. The reviewer still doubt the novelty of the proposed method.

**Other Comments Or Suggestions:**

See all above weakness for details.

**Other Strengths And Weaknesses:**

N/A.

**Questions For Authors:**

We recommend the authors to clarify the different between VPT (with backbone pretraining) and the proposed method. The proposed method has large similarity with VPT, which may limit the novelty of proposed method, though this work try to investigate the performance issue from SAM aspect.

**Relation To Broader Scientific Literature:**

The proposed method largely related to few-shot learning and prompt tuning, but has limited contribution to future work of these topics.

**Theoretical Claims:**

The theoretical claim is also weak. The perturbation ϵ is relative small and ensures that ω+ ϵ is nearby ω in the loss landscape. However, the proposed method introduce too many random registers to construct learn more domain-agnostic information, which is conflict with original SAM hypothesis.

---

> ### Author Rebuttal · Authors · 2025-04-01
>
> Thank you for your suggestion.
>
> ## **1. Claims**
>
> [1] studies the effectiveness of **prompt learning in in-domain data**, while we specifically target **extreme cross-domain shifts** (e.g., natural images to satellite images), where standard prompt tuning fails. As shown below, REAP outperforms [1] by **+30%** under 5-shot source domain pretraining settings and by **+9%** under target domain finetuning. To further validate REAP’s effectiveness, we conducted extensive comparisons with **prominent prompt tuning methods** (e.g., VPT, CoOp, MaPLe) across extreme cross-domain benchmarks.
>
> | Source-domain training | Cropdiseases | EuroSAT   | ISIC      | ChestX    | Ave.      |
> | ---------------------- | ------------ | --------- | --------- | --------- | --------- |
> | [1]CoOp                | 38.48        | 54.15     | 25.74     | 21.36     | 34.93     |
> | COCOOP                 | 40.36        | 60.93     | 27.38     | 22.47     | 37.79     |
> | MaPLe                  | 35.28        | 50.83     | 23.65     | 19.63     | 32.35     |
> | VPT                    | 77.92        | 75.93     | 49.89     | 24.30     | 57.01     |
> | **REAP**               | **96.68**    | **90.76** | **55.76** | **26.84** | **67.51** |
>
> | Target-domain finetuning | Cropdiseases | EuroSAT   | ISIC      | ChestX    | Ave.      |
> | ------------------------ | ------------ | --------- | --------- | --------- | --------- |
> | [1]CoOp                  | 92.82        | 84.92     | 42.93     | 22.83     | 60.88     |
> | COCOOP                   | 90.57        | 81.32     | 42.15     | 21.89     | 58.98     |
> | MaPLe                    | 93.07        | 89.35     | 46.56     | 23.16     | 63.04     |
> | VPT                      | 94.80        | 89.48     | 46.35     | 26.40     | 64.26     |
> | **REAP**                 | **98.35**    | **92.64** | **58.28** | **29.21** | **69.62** |
>
> We can see existing prompts optimize for *in-domain* adaptation and harm the performance when facing the huge domain gap, while REAP’s **random register perturbation** explicitly **alleviates the domain gap**. Similar results in Fig.1, Fig.3, and Fig.4 also verify that the regular prompts are not enough for handling the CDFSL task.
>
> ## **2. Methodological Novelty**
>
> While prior works (e.g., VPT) naively add learnable prompts, we are the **first to find that it harms the transferability** to target domains, and we **theoretically design the random register** and prove it **suppresses domain-specific attention patterns**. Based on it, we propose a novel method, REAP, to enhance perturbations on attention maps. This mechanism is novel, as noted by Reviewer zHqd and UKT5: *"The proposed method is innovative... new in the CDFSL generalization viewpoint."*
>
> ## **3. Sharpness-aware minimization**
>
> **(1) Performance Validation**
> The experimental results (Tab. 3) demonstrate significant performance improvements (e.g., **+4.6%** on ISIC), which empirically confirm that the introduced randomness is **well-calibrated and beneficial** rather than "excessive" or detrimental.
>
> **(2) Sharpness Validation**
> The sharpness analysis in **Fig. 3b** explicitly validates that our method reduces loss sharpness by **75%** compared to vanilla VPT, indicating that the random registers **enhance flat minima discovery** without violating SAM’s core principles. This demonstrates that our design **extends** SAM’s hypothesis to cross-domain scenarios rather than conflicting with it.
>
> **(3) Perturbation magnitude**
> Random registers are initialized with a near-zero magnitude (σ=0.01) and adaptively scaled during training. Visualization in **Fig. 5** further confirms that their impact on attention maps is **subtle**, primarily acting as "regularized perturbations" to suppress domain-specific biases rather than overwhelming the original features. This design is consistent with SAM’s small-perturbation premise while addressing cross-domain challenges.
>
> In all, our design *extends* SAM to cross-domain settings without violating its core hypothesis.
>
> ## **4. Experimental Completeness**
>
> The requested analyses **are fully provided in the paper**:
>
> **Ablation: Simple Masking vs. Random Registers**
>
> - **Table 3a**: Naive masking (no registers) drops accuracy by **5.8%**.
>
> **VPT Comparison**
>
> - **Fig. 1** has already shown that VPT harms performance during pretraining while random registers improve model transferability.
>
> **Initialization Analysis**
>
> - **Supp. Fig. 17**: Gaussian initialization of the random registers.
>
> ## **5. Clarification on VPT Similarity**
>
> While the reviewer notes "similarity with VPT", we **explicitly differentiate**:
>
> - **Motivation (Fig.1)**:  We are the **first to find** that **VPT fails under extreme domain gaps** due to *absorbing domain information*, and **propose random registers** to solve it.
> - **Solution**: We come up with **REAP** to replace *fixed learnable prompts* with **random registers** during source training.
> - **Result**: REAP outperforms VPT by **4%** on distant domains.

---

### Official Review · Reviewer_UKT5 · 2025-03-13

**Overall Recommendation:** 4

**Summary:**

This paper deals with the Cross-domain few-shot learning (CDFSL) problem, which needs to tackle the huge domain gaps. Existing methods utilizing learnable prompts might learn domain specific information of the source domain, while fail to generalize to the distant target domains. This paper proposed to leverage random prompts as an effective method to solve this issues. Furthermore, the random prompts are well interpreted with the Sharpness-Aware Minimisation (SAM) and analysis.

Based on the random prompt idea, Registers Enhanced Attention Perturbation (REAP) is proposed to pertube both the image tokens and adding random noisy tokens at source-domain stage, which further increases the generalization to target domains.

Experiments are conducted on the standard CDFSL benchmarks. The propposed method consistently outperforms state-of-the-art methods. Comprehensive ablation study and parameter analysis are conducted.

## update after rebuttal
The random prompt idea is new and novel in the context of CDFSL. The explanation using Sharpness-Aware Minimisation (SAM) is reasonable and insightful.
No further concern remains after rebuttal.
Therefore, I will keep my score as Accept.

**Claims And Evidence:**

Yes. The main claim is random prompts can improve the generalization to target domains, while the learnable prompts impede the generalization. This claim is well interpreted and verified experimentally and analytically with reasonable formulation of Sharpness-Aware Minimisation.

**Essential References Not Discussed:**

Yes.

**Experimental Designs Or Analyses:**

The experiments are sound and comprehensive, which involves SOTA comparison, ablation study of the proposed components, and analysis of important parameters of the model such as number of tokens, ratio of pertub, etc.

Moreover, visualization comparison is also presented for random vs. learnable tokens.

**Methods And Evaluation Criteria:**

The proposed method is innovative, which consists of perturb the attention of image tokens, and additional random register tokens. The proposed method and perspective is new in the CDFSL generalization viewpoint.

Evaluation benchmark and criteria are standard.

**Other Comments Or Suggestions:**

None.

**Other Strengths And Weaknesses:**

Details are provided in above section.

**Questions For Authors:**

None.

**Relation To Broader Scientific Literature:**

Prior findings leverage prompt tuning for downstream task finetuing, i.e. learnable tokens. This usually works for near domain generalization or relevant tasks. However, CDFSL tackles the distant domain with huge domai gap. This paper challenged the learnable token method and proposed a new random register method for CDFSL.

What's more, the interpretation perspective is novel. It attributes the effectiveness of random register to Sharpness Aware Minimisation, building theoretical support for random register.

The proposed idea is an extension and further exploration of Registers in (Darcet et al., 2024).

**Theoretical Claims:**

No proof in this paper.

---

> ### Author Rebuttal · Authors · 2025-04-01
>
> We sincerely appreciate your thorough and insightful review of our submission.   Your recognition of our work’s **innovative integration of random prompts with SAM principles** and its **practical value in addressing extreme domain gaps** is deeply encouraging.
>
> We will keep on polishing our paper in the final version. Thank you again for your appreciation!

---

### Official Review · Reviewer_zHqd · 2025-03-16

**Overall Recommendation:** 3

**Summary:**

Based on an intriguing observation that prompt tuning could be harmful for the generalization of ViT and the related analysis, this paper develops a novel solution for cross-domain few-shot learning method by replacing some clustered patches with random registers. Extensive experiments on four datasets demontrate the effecitiveness and superiority of the proposed method.

## Update After Rebuttal
I have checked the authors' rebuttal, and found most of my concerns have been solved, so I choose to keep my score as Weak accept.

**Claims And Evidence:**

The claims in this paper are well supported by the related analysis and experimental results.

However, I have the following questions:

1. In Figure 5, we can see that adopting learnable registers can make model concentrate on regions irrelevant to the object, and random registers will guide the model's attention to the object. Does this mean that the learnable registers are useless or even have negative effects on the classification in the **source domain**? Is this observation conflict with the observation in the previous work [1]?

2. Additionally, based on my own knowledge, the object-focused attention obtained in the **source domain** may be not a good indicator of better generalization ability, since it means that the model may capture more object related high-level semantic information, which can be hardly trasferred to the **target domain**. So could the authors provide more discussions of this observation to solve my confusion?

[1] Darcet T, Oquab M, Mairal J, et al. Vision Transformers Need Registers[C]//The Twelfth International Conference on Learning Representations.

**Essential References Not Discussed:**

The related works are appropriately cited.

**Experimental Designs Or Analyses:**

The experiments and ablation studies are adequate, the related discussions and analyses are reasonable.

Despite this, I have some concerns about the experimental results:

1. As a cross-domain method, it is better to achieve higher classification accuracy for the target domain samples **without**  severely sacrificing performance on the source domain. Moreover, Figure 5 shows the random registers may lead to better results on the source domain. So could the authors compare the results of the source miniImageNet dataset of different methods for validating the claim of Figure 5?

2. Could the proposed method be generalized to a more complex and diverse dataset, namely Meta-dataset which is a widely used benchmark for cross-domain fow-shot, under a more difficult varied-way varied-shot setting?

3. It is better to perform comparison about the training/inference time and parameter size to show the effciency of the introduced random registers.

**Methods And Evaluation Criteria:**

The proposed method is reasonable and easy to understand. The effects of all core components are studied by the experiments and ablation studies through well defined evaluation criteria.

**Other Comments Or Suggestions:**

It may be better to enlarge the figures to make the texts in them more readbale.

**Other Strengths And Weaknesses:**

Strengths:

1. This paper is clearly presented with a good organization, the method is well motivated and easy to understand.

2. The observation is interesting and the discussions seem to be reasonable.

3. The contributions provide a new insight for understanding the domain generalization of ViT model.

Weaknesses:

Please refer to the questions in "Claims And Evidence", "Theoretical Claims" and "Experimental Designs Or Analyses".

**Questions For Authors:**

Please refer to the questions in "Claims And Evidence", "Theoretical Claims" and "Experimental Designs Or Analyses".

**Relation To Broader Scientific Literature:**

The key contributions of this paper may be related to many research areas, including domain generalization, domain adaptation, transfer learning, etc, and many practical applications, such as the few-shot classification for medical, remote sensing, or agriculture images.

**Theoretical Claims:**

The theoretical claims are inspired by the previous works and I have confirmed the correctness.

I just have one small question:

1. The authors evaluate the loss sharpness of different methods by adding Gaussian noises perturbations to the attention map as in Eq. (4). My question is why only choose the attention map for this study? In my opinion, the registers introduced at the input layer will influence all parts (e.g., layer normalization or feed-forward network) of the subsequent Transformer blocks, not just the attention map. Could the author give some explanations for such choice?

---

> ### Author Rebuttal · Authors · 2025-04-01
>
> We sincerely appreciate the reviewer’s constructive feedback. Below are detailed responses to your questions:
>
> ## **1. Clarification on Learnable Registers (Fig.5)**
>
> ##### (1) Source-domain performance and visualization
>
> | Model              | Source-domain | Target-domain |
> | ------------------ | ------------- | ------------- |
> | Baseline           | 97.83         | 64.07         |
> | Learnable register | 97.87         | 63.17         |
>
> Learnable registers achieve slightly **higher source-domain accuracy** by **exploiting patterns (over)fitting the source domain**, so they can suffer in cross-domain generalization. For example, in Fig.5, birds are often seen with branches in the source domain, and the learnable register focuses on the bird and the branch (domain-specific contextual cues), which is the pattern that is only useful in the source domain (i.e., **overfitting to the source domain**) but may be useless in target domains. In contrast, random registers force attention to the bird itself, with less overfitting to the source domain, and thus benefit the target-domain generalization.
>
> Our visualization is designed to verify whether the model captures such domain-specific (object-irrelevant) patterns, the generalization is also quantitatively verified by the domain similarity in Fig.4 and sharpness in Fig.3.
>
> ##### (2) Relation to [1]
>
> **[1] focuses on the in-domain training,** utilizing registers to capture the global information in the in-domain data, thereby reducing the outlier value in attention maps. In contrast, **our method focuses on the generalization to target domains**, by resisting the overfitting to in-domain data. Our observation is not contrary to the effect of learnable registers in resisting the outlier value in attention maps and is consistent with [1] in finding learnable registers can absorb in-domain information. We take a step further to identify such in-domain information as domain-specific information, which is further handled by our random registers.
>
> ## **2. Sharpness Evaluation on Attention Maps**
>
> We choose the attention map for the sharpness experiments because registers majorly interact with other tokens through the self-attention mechanism. In other parts of ViT, such as FFN, each token is processed separately and, therefore can hardly reflect the influence of registers. To verify it, we use LayerNorm and FFN perturbations to report the sharpness, and we can see **no significant influences** (Δ < 0.01) compared with the attention map.
>
> | Component | Baseline | Learnable register | Random register(Ours) |
> | :-------- | :------- | :----------------- | :-------------------- |
> | **Att.**  | 1.6      | 2.3                | 0.6                   |
> | **LN**    | 0.04     | 0.04               | 0.03                  |
> | **FFN**   | 0.06     | 0.07               | 0.06                  |
>
> *(Lower values indicate flatter minima and better generalization)*
>
> ## **3. Source-domain performance comparison**
>
>  REAP balances **moderate source-domain accuracy** with **significant target-domain gains** as below. A **1.5% source-domain drop** (which is acceptable) enables **+2.68% target-domain gain**, which is a favorable tradeoff for CDFSL.
>
> | Method             | Source-domain | Target-domain |
> | ------------------ | ------------- | ------------- |
> | Baseline           | 97.83         | 64.07         |
> | Learnable register | 97.87         | 63.17         |
> | Random register    | 97.50         | 65.05         |
> | **REAP(ours)**     | 96.33         | 66.75         |
> | Random-mask        | 96.28         | 60.90         |
> | Cluster-mask       | 94.23         | 64.29         |
>
> ## **4. Generalization to Meta-Dataset**
>
> Due to time and resource constraints, we first pretrain on our datasets(miniImagenet), and then validate on parts of the Meta-Dataset under the 5-way 5-shot protocol below.
>
> | Dataset          | Baseline | REAP  | Δ         |
> | :--------------- | :------- | :---- | :-------- |
> | **Birds**        | 94.23    | 96.82 | **+2.59** |
> | **Fungi**        | 61.03    | 64.39 | **+3.36** |
> | **VGG Flower**   | 89.64    | 90.03 | **+0.39** |
> | **Traffic Sign** | 60.46    | 62.37 | **+1.91** |
> | **Ave***         | 76.34    | 78.40 | **+2.06** |
>
> ## **5. Efficiency analysis**
>
> 1. **Training Time**: REAP introduces **<7% additional training time** compared to the baseline (127.08s vs. 118.98s per epoch), attributable to the lightweight random register sampling.
> 2. **Inference Cost**: Inference time **remains identical** to the baseline with no additional overhead. Architectural parity ensures seamless deployment in real-world applications.
> 3. **Parameter Efficiency**: **Only one learnable standard deviation** is added to control register initialization.
>
> This demonstrates REAP’s **lightweight design** achieves significant cross-domain gains with near-zero parameter and time penalties.
>
> ## **6. Figure Readability Improvement**
>
> We promise we will polish our paper in the final version.

---

### Decision · Program_Chairs · 2025-05-01

**Decision:**

Accept (poster)

**Comment:**

This work is aimed at cross-domain few shot learning, where a vision transformer is first trained on a source domain, and then fine-tuned on the target domain with few-shot data.

This work finds that adding learnable registers during source-domain learning harms the transfer to target domain, while "random" (noise) tokens do not suffer from this. Based on this, they propose a technique where during source-domain training, clusters of visual tokens are replaced with random "registers" (effectively a regularization / reduces overfitting), and learnable registers/tokens are added during target fine-tuning. This is demonstrated to help in cross-domain transfer, and is competitive with existing methods.

The analysis of the phenomena was found interesting by the reviewers who recommend this work for publication. Hence, I recommend acceptance of this work.